# Drug Repositioning in Intensive Care Patients and Pharmacokinetic Variability: The Illustration of Hydroxychloroquine

Gwendoline Ragonnet [1,*], Elisabeth Jouve [1], Lionel Velly [2], Marc Leone [3], Gary Duclos [3], Jeremy Bourenne [4], Karim Harti Souab [5], Caroline Solas [6] and Romain Guilhaumou [1]

[1] Service de Pharmacologie Clinique et Pharmacovigilance, Institut des Neurosciences des Systèmes, Inserm UMR 11600, Aix Marseille University APHM, CEDEX 5, 13385 Marseille, France; elisabeth.jouve@ap-hm.fr (E.J.); romain.guilhaumou@ap-hm.fr (R.G.)

[2] Department of Anesthesiology and Critical Care Medicine, Timone University Hospital, Aix Marseille University APHM, 13005 Marseille, France; lionel.velly@ap-hm.fr

[3] Department of Anesthesiology and Intensive Care, Timone University Hospital, Aix Marseille University APHM, 13015 Marseille, France; marc.leone@ap-hm.fr (M.L.); gary.duclos@ap-hm.fr (G.D.)

[4] Emergency Resuscitation Department, Timone University Hospital, Aix Marseille University APHM, CEDEX 5, 13385 Marseille, France; jeremy.bourenne@ap-hm.fr

[5] Intensive Care Unit, Timone University Hospital, Aix Marseille University APHM, CEDEX 5, 13385 Marseille, France; karim.harti-souab@ap-hm.fr

[6] Unité des Virus Émergents IRD190, Inserm 1207, Laboratoire de Pharmacocinétique et Toxicologie, CEDEX 5, 13385 Marseille, France; caroline.solas@ap-hm.fr

[*] Correspondence: gwendoline.ragonnet@ap-hm.fr

**Abstract:** During the SARS-CoV-2 pandemic, hydroxychloroquine (HCQ), was among the first drugs to be tested due to demonstrated in vitro antiviral activity against SARS-CoV-2. Pharmacokinetic variability was expected due to the frequent comorbidities and pathophysiological modifications observed in severe COVID-19 patients hospitalized in intensive care units (ICUs). The aim of this study was to describe HCQ plasmatic concentrations in ICUs and assess variability factors. A multicentric retrospective study was carried in four ICUs in Marseille from March to April 2020. There were two dosing regimens: 400 mg after a 400 mg loading dose (DR1); and 600 mg without a loading dose (DR2). HCQ concentrations were determined every 2 or 3 days. The impacts of demo-graphic, biological, and clinical covariates were investigated. The median HCQ concentration was: 0.096 mg/L on day (D) 2, 0.129 mg/L on D3 to D5, 0.140 mg/L on D6 to D10 for DR1 versus 0.116 mg/L, 0.261 mg/L, and 0.30 mg/L, respectively, for DR2. At D2, 53.9% and 46.2% of patients with DR1 and DR2, respectively, presented HCQP concentrations <0.1 µg/mL and 48.2% versus 10.7% at D3 to D5. Time post-initiation, dosing regimen, nasogastric administration, and weight showed significant association with HCQ variability. The high proportion of suboptimal HCQ concentrations can be explained by a lack of optimized dosing regimen and numerous pathophysiological changes in the COVID-19/ICU population.

**Keywords:** hydroxychloroquine; intensive care unit; COVID-19; pharmacokinetics; drug repositioning

## 1. Introduction

First reported in China in December 2019, the COVID-19 epidemic, causing severe acute respiratory syndrome, is ongoing. The World Health Organization described the global COVID-19 situation as a pandemic on 11 March 2020. In this context, several drugs have been repurposed as potential candidates for the treatment of SARS-CoV-2 infection. Preliminary choices were essentially based on in vitro potency and clinical translation in-to effective therapies and may be challenging due to in vivo pharmacokinetic (PK) and pharmacodynamic (PD) properties [1]. Hydroxychloroquine (HCQ), a well-known drug

effective in the treatment of malaria and autoimmune diseases [2–4], was among the first drugs to be tested due to demonstrated in vitro antiviral activity against SARS-CoV-2.

Approximately 25% of hospitalized patients infected by SARS-CoV-2 required intensive care unit (ICU) admission, and HCQ was widely prescribed in this population [5]. Indeed, numerous studies have demonstrated that antimicrobial plasma concentrations are often variable and unpredictable in this population [6]. Due to the frequent comorbidities (i.e., obesity, diabetes, and cardiovascular complications) and pathophysiological modifications (cytokine storms, multivisceral failure, and life-threatening prognosis) observed in severe COVID-19 patients, high antimicrobial PK variability was expected in this population.

Moreover, HCQ presents uncommon PK properties, with strong tissue tropism (in particular for the kidneys and liver) and a long half-life [7], combined with a narrow therapeutic index [8]. In this context, HCQ therapeutic drug monitoring (TDM) was proposed in the French Pharmacology Committee guidelines [9]. Despite there being no clear association between concentration and response and/or side effects [10,11], a minimal plasma threshold of 0.1 mg/L was proposed based on in vitro and modelling experiments [12,13].

Little information was available concerning HCQ pharmacokinetics in ICU patients, especially in the context of COVID-19, and the PK parameters of HCQ were mostly estimated from studies carried out on patients with rheumatoid arthritis or lupus [14,15] or on healthy patients [16]. Moreover, the variability of HCQ patient exposure may be in-creased by the lack of consensus on the optimal dosing regimen. Indeed, several dosing regimens have been proposed, based on a PD/PK model and simulations or adapted from other indications [17]. In this context, the aim of this study was to describe HCQ plasma concentrations in ICU patients and assess variability factors.

## 2. Materials and Methods

A multicentric, retrospective study was carried out in four ICUs in Marseille University Hospitals on hospitalized patients from March 20 to 13 April 2020. Patients >18 years of age, infected by COVID-19, treated with HCQ, and with at least one available plasma concentration measurement were eligible for inclusion. Two different dosing regimens were used according to ICU protocols: 400 mg daily (200 mg twice a day (BID) or 400 mg once a day (QD)) after a 400 mg BID loading dose (dosing regimen 1, DR1) [12] or 200 mg three times a day (TID) without a loading dose (dosing regimen 2, DR2) [17]. For patients under mechanical ventilation, the 200 mg film-coated tablet was ground and administrated by an enteral feeding tube (EFT).

This study was declared to the local institutional committee as a retrospective, non-interventional, and unnamed study (AP-HM N° PADS20-200, 20 April 2020).

Plasma HCQ concentrations were determined using ultra-high-performance liquid chromatography–tandem mass spectrometry (UPLC-MS/MS, Waters, Milford, MA, USA) with a lower limit of quantification of 0.015 mg/L [18]. The assay matrix was plasma, the exact time of blood sampling was recorded by the medical staff, and samples were immediately sent to the laboratory and aliquoted into 2 mL polypropylene tubes. As recommended, plasma samples for HCQ determination were collected at multiple time points: 48 h post-treatment initiation and every 2 or 3 days until the end of treatment. TDM was performed before steady state was reached, given the long half-life of HCQ and the need for short and rapidly effective treatment [8]. For patients with HCQ plasma concentrations below the 0.1 mg/L thresh-old, the dose could be increased, given the clinical context.

A descriptive study was performed examining plasma HCQ concentrations according to the dosing regimen and timing. Patient characteristics are presented as the frequency and percentage of patients for categorical variables and by the mean and standard deviation (SD) for continuous variables. Differences between the two dosing regimens were tested using the Student's *t*-test and the chi-square test (or Fisher exact test). HCQ concentrations

are presented as the mean, SD, and coefficient of variation (%) and were compared between the two dosing regimens according to the time from treatment initiation: day 2, days 3–5, and days 6–10. Linear mixed models with random intercepts and subjects were used to explore the association between clinical and biological parameters and HCQ concentration. The quantitative covariates studied were age, weight, body mass index (BMI), creatinine concentration, and estimated glomerular filtration rate (eGFR: calculated with the CKD-EPI formula), as well as urea, C reactive protein (CRP), potassium, and albumin concentrations. The qualitative covariates were sex, administration by EFT, moderate-to-severe renal failure (eGFR < 60 mL/min), weight > 90 kg, and BMI > 30 kg/m$^2$. Biological and clinical data were recorded concurrently with the TDM sample. The retained model is presented as coefficients with 95% confidence intervals (CI). Statistical significance was considered for $p \leq 0.05$. Calculations were performed using SAS (version 9.4, SAS Institute Inc., Cary, NC, USA).

## 3. Results

In total, 139 plasma samples were collected from 76 ICU patients, 40 treated with DR1 and 36 with DR2. The baseline demographics and clinical characteristics, depending on the dosing regimen, are presented in Table 1. The patients were mostly men, between 60 and 70 years of age, with a BMI > 30 kg/m$^2$. The two populations were comparable, except for age, eGFR, and CRP levels.

**Table 1.** Patient characteristics. (n (%) or median $\pm$ standard deviation).

|  | DR1 * | DR2 ** | *p*-Value |
|---|---|---|---|
| Number of patients | 40 | 36 | - |
| Number of samples | 67 | 72 | - |
| Age, years | 67.5 $\pm$ 8.3 | 59.5 $\pm$ 12.3 | <0.0001 |
| Sex, M | 54 (80.6) | 57 (79.2) | 0.834 |
| Weight, kg | 83.4 $\pm$ 16.0 [1] | 88.4 $\pm$ 18.1 | 0.122 |
| Weight < 90 kg | 19 (37.3) | 26 (36.1) | 0.897 |
| BMI, kg/m$^2$ | 29.5 $\pm$ 6.2 [2] | 29.6 $\pm$ 6.1 | 0.909 |
| BMI < 30 kg/m$^2$ | 11 (45.8) | 26 (36.1) | 0.397 |
| Mechanical ventilation | 49 (73.1) | 51 (72.7) | 0.971 |
| eGFR, mL/min | 78.3 $\pm$ 28.0 [3] | 92.2 $\pm$ 30.3 [4] | 0.010 |
| Moderate-to-severe kidney failure (GFR < 60) | 10 (16.4) | 10 (17.0) | 0.935 |
| Creatinine, μmol/L | 110 $\pm$ 114 | 86.9 $\pm$ 58.1 | 0.147 |
| Urea, mmol/L | 10.4 $\pm$ 7.3 | 9.7 $\pm$ 6.0 | 0.570 |
| CRP, mg/L | 167 $\pm$ 104 [5] | 213 $\pm$ 101 [6] | 0.024 |
| CRP > 100, mg/L | 33 (68.8) | 47 (81.0) | 0.143 |
| K$^+$, mmol/L | 4.3 $\pm$ 0.6 | 4.2 $\pm$ 0.8 | 0.321 |
| Albumin, g/L | 26.6 $\pm$ 25.7 [7] | 26.2 $\pm$ 3.8 | 0.560 |
| QT prolongation, ms | 5 (7.5) | 3 (4.2) | 0.482 |

* DR1: dosing regimen 1—400 mg daily (200 mg BID or 400 mg QD) after a 400 mg BID loading dose. ** DR2: dosing regimen 2—200 mg TID; [1] n = 51, [2] n = 24, [3] n = 61, [4] n = 59, [5] n = 48, [6] n = 58, [7] n = 66.

The median HCQ concentrations observed according to the dosing regimen were (standard deviation, CV%): 0.096 mg/L on day 2 ($\pm$0.052, 54%), 0.129 mg/L on days 3–5 ($\pm$0.078, 61%), and 0.140 mg/L ($\pm$0.072, 50%) on days 6–10 for DR1 and 0.116 mg/L ($\pm$0.065, 56%) on day 2, 0.261 mg/L on days 3–5 ($\pm$0.090, 46%), and 0.30 mg/L ($\pm$0.082, 31%) on days 6–10 for DR2 (Figure 1).

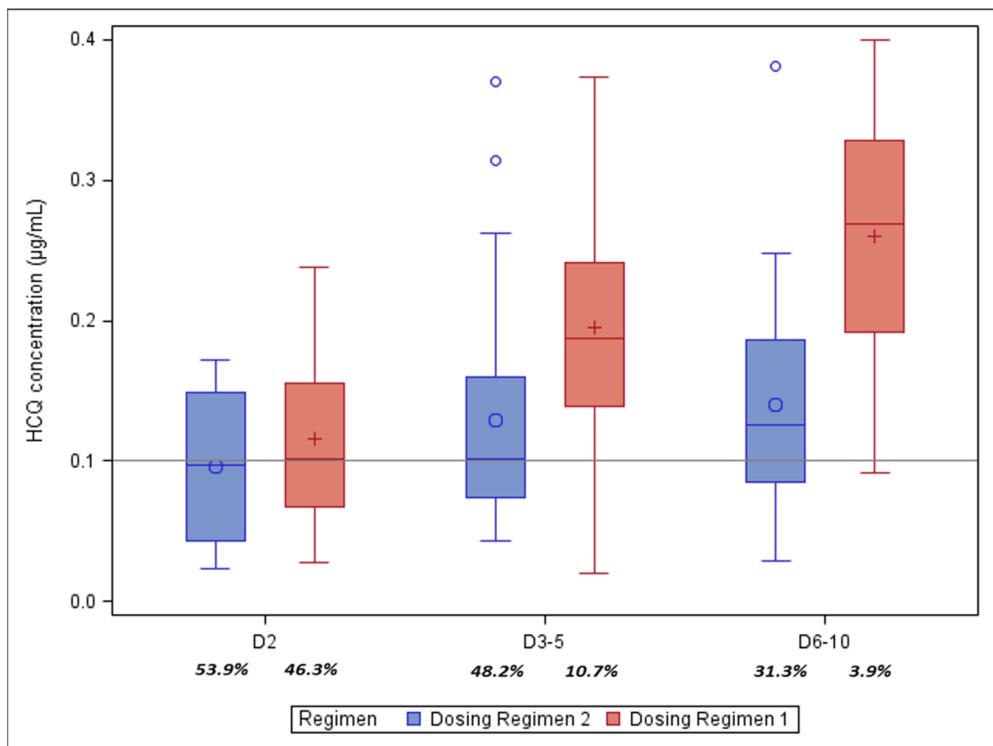

**Figure 1.** HCQ concentrations according to the time from treatment initiation and dosing regimen. % in italics indicates the % of concentrations <0.1 µg/mL.

On day 2, 53.9% and 46.2% of patients treated with DR1 and DR2, respectively, presented HCQ plasma concentrations <0.1 µg/mL, 48.2% and 10.7% on days 3–5, and 31.3% and 3.9% on days 6–10. Among the patients treated with DR1, the daily dose was increased for seven (17.5%): two between days 1 and 2 (600 mg), three between days 3–5 (800 mg), and two between days 6–10 (800 mg). Among the patients treated with DR2, the daily dose was increased for seven (19.4%): three between days 3–5 (1000 mg, n = 1; 1200 mg, n = 2) and four between days 6–10 (1000 mg).

We studied the impact of ICU patient covariables on circulating HCQ concentrations by first performing univariate analysis. Only variables with a *p*-value < 0.2 were retained for multivariate analysis: time post-initiation (*p* = 0.0005), dosing regimen (*p* = 0.0015), CRP (*p* = 0.0791), moderate-to-severe renal failure (*p* = 0.1573), EFT administration (*p* = 0.0101), and weight (*p* = 0.0288). In multivariate analysis, time post-initiation (*p* = 0.001), dosing regimen (*p* = 0.0029), EFT administration (*p* = 0.0081), and weight (*p* = 0.0333) showed a significant association with HCQ concentration variability.

## 4. Discussion

The main results of our study can be summarized as follows. (1) High variability in HCQ plasma concentrations was observed in ICU COVID-19 patients. (2) Time from treatment initiation, dosing regimen, administration by EFT, and weight were identified as the main factors of HCQ exposure variability. (3) The use of a loading dose and high daily dosage is required to optimize HCQ exposure.

Patients included in the present study were representative of the ICU population hospitalized for severe forms of SARS-CoV-2 infection between March and April 2020 in France: an elderly population (64.5% ≥ 60 years of age), predominantly men (79.9%), overweight (71.7%), under mechanical ventilation (67%), and presenting serious inflammatory syndrome [19,20]. As expected in this specific population, HCQ plasma concentrations were highly variable.

We identified delay post-administration and daily dosing as variability factors of HCQ concentration. The goal with SARS-CoV2-infected patients was to quickly reduce the viral load (as HCQ was proposed as an antiviral), which is problematic for a drug with a long half-life for which the steady state will not be reached before the end of treatment. These results highlight the importance of determining an optimized dosing regimen in drug repositioning, notably in acute, severe infection. First, the requirement of a loading dose merits consideration. In the present study, HCQ concentrations at day 2 were no different between the two dosing regimens, despite lower daily dosing for DR1 (400 mg vs. 600 mg). These results support the value of a loading dose to optimize exposure in the first days of treatment, as previously described by Lê et al. in the DisCoVeRy trial [20]. Despite an 800 mg loading dose at D0, more than 50% of patients had suboptimal plasma HCQ concentrations in our population. A larger loading dose would have been more efficient to quickly reach the 0.1 mg/L threshold. However, the loading dose is not sufficient to maintain HCQ plasma exposure throughout treatment, and an appropriate daily dose should be considered. Our results indicate that a higher dose is necessary to optimize dose exposure. Indeed, patients treated with DR2 had higher HCQ concentrations and presented a better target attainment after the third day of treatment than patients treated with DR1. The combination of a higher loading dose and higher daily dosing was surely the best choice to maintain HCQ concentrations above the 0.1 mg/L plasma threshold.

We also identified administration by EFT and weight as variability factors. Lower HCQ concentrations were observed for patients under mechanical ventilation and/or those who were overweight. During HCQ administration via EFT, the HCQ film-coated tablets were ground up, and absorption was probably reduced. Such PK variability has been previously described for other drugs [21], and a higher dosing regimen should be considered for such patients. Weight was also identified as a variability factor of HCQ clearance in two recent population PK models [22]. Overweight patients presented higher HCQ clearance and therefore lower exposition. A dose-adjusted to body weight should thus be considered to optimize HCQ treatment in the ICU population. The occurrence of an inflammatory reaction may result in changes in the PK of a drug by inhibiting its intrinsic clearance by inhibiting hepatic cytochromes or altering its binding to plasma proteins for strongly bound drugs. An impact of inflammation on PK has been highlighted for other anti-infective drugs in COVID-19 patients [21–24] and in other diseases [25]. However, we observed no impact of CRP variability on HCQ concentrations, probably due to the mixed urinary and hepatic elimination of this drug, in contrast to lopinavir or voriconazole, which are exclusively metabolized by the hepatic route [26].

Based on in vitro HCQ potency and clinical translation, in the complete absence of drugs with proven efficacy against COVID-19, hydroxychloroquine has been used off-label by many physicians worldwide as a potential anti-COVID-19 drug. A retrospective analysis that reported an increased risk of serious heart disorders in patients treated with HCQ was published in the Lancet journal on 22nd May. The Lancet paper was then retracted over the authenticity of the unreleased patient database. In the meantime, the Food and Drug Administration (FDA) had revoked the Emergency Use Authorization (EUA), the World Health Organization halted its trial of hydroxychloroquine in hospitalized COVID-19 patients, and the European Medicine Agency stopped the off-label use of hydroxychloroquine. Many publications have been made so far on the use of hydroxychloroquine for treatment or prevention of COVID-19, but only few have come from highly ranked published studies, and there is still a need for further evidence on the benefits and risks of hydroxychloroquine in different settings. The retrospective, observational design of our study is its main limitation. First, it was not possible to record all covariate data and information on weight, GFR, CRP, and/or albumin, which were lacking for some patients. In addition, there was a selection bias between patients treated with the two dosing regimens. Patients treated with DR2 had higher CRP concentrations, and those treated with DR1 were older, with a lower median eGFR. However, CRP, age, and eGFR were not identified as variability factors of HCQ concentrations in our study. An influence of renal function on HCQ PK was

previously described but limited to severe impairment [27], and the difference between the two groups was no longer observed considering severe-to-moderate renal failure. Finally, the correlation of observed pharmacokinetic variability and clinical outcomes was not studied. Indeed, the clinical effectiveness of hydroxychloroquine against COVID19 has not been demonstrated in randomized clinical trials, and the European Medicine Agency stopped the off-label use of hydroxychloroquine.

## 5. Conclusions

In conclusion, this study illustrates the large interindividual variability of HCQ concentrations in the ICU population. The high proportion of suboptimal plasma concentrations can be explained by the lack of an optimized dosing regimen and the numerous pathophysiological changes observed in the combined COVID-19/ICU population. Although the efficacy of HCQ was not confirmed in COVID-19 patients [28], this study high-lights the requirement to characterize drug exposure and variability factors in the context of drug repositioning, especially in the ICU population.

**Author Contributions:** Conceptualization, G.R., C.S. and R.G.; methodology, G.R., C.S. and R.G.; software, G.R. and E.J.; validation, G.R., C.S. and R.G.; formal analysis, G.R., E.J. and R.G.; investigation, G.R., L.V., M.L., G.D., J.B. and K.H.S.; resources, L.V., M.L., G.D., J.B. and K.H.S.; data curation, G.R. and E.J.; writing—original draft preparation, G.R.; writing—review and editing, G.R.; visualization, G.R. and R.G.; supervision, C.S.; project administration, C.S. All authors have read and agreed to the published version of the manuscript.

**Funding:** This research received no external funding.

**Institutional Review Board Statement:** The study was conducted according to the guidelines of the Declaration of Helsinki and approved by the Institutional Review Board, AP-HM N° PADS20-200 (20 April 2020).

**Informed Consent Statement:** Patient consent was waived due to a retrospective, non-interventional and unnamed study.

**Data Availability Statement:** Not applicable.

**Conflicts of Interest:** The authors declare no conflict of interest.

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
