# Peer review of "Drug Repositioning in Intensive Care Patients and Pharmacokinetic Variability: The Illustration of Hydroxychloroquine"

_futurepharmacol, doi:10.3390/futurepharmacol2010007_

Round 1

Reviewer 1 Report

The topic of the manuscript is interesting and the reviewer feels it can be accepted after some minor amendments.

1) This is a communication. Why the abstract is so long?

2) Why the age between DR1 and DR 2 was different?

3) The race should be indicated in Table 1.

4) Is the clinical outcome (e.g. survival rate) correlated with the variability?

Author Response

Respond to reviewers

 Reviewers 1.

1) This is a communication. Why the abstract is so long?

According to the instructions to authors, the abstract have a total of 200 words.

2) Why the age between DR1 and DR2 was different?

This is a selection bias, which is one the the main weakness of a retrospective study. Of the 4 intensive care units studied, two used DR1 and two others DR2. The dosing regimen administered was not dependent on age.

3) The race should be indicated in Table 1.

Present study was a retrospective, unnamed study and authors have not access to race data in the records.

4) Is the clinical outcome (e.g. survival rate) correlated with the variability?

We thank the reviewer for this relevant comment. It would have been interesting to study the correlation between clinical outcomes and concentrations variability. Unfortunately, data on clinical outcomes were not available. Moreover, hydroxychloroquine clinical effectiveness against COVID19 has not been demonstrated in randomized clinical trial. This point has been added in the limitation section, please see the attachment. .

Reviewer 2 Report

Ragonnet et al describe the pharmacokinetics of hydroxychloroquine usage in ICU for COVID-19 patients. The study is interesting and well written. However, i shall suggest few points to discuss/change.

1) Authors only discuss the pharmacokinetics of different doses of hydroxychloroquine which is largely descriptive. I would suggest to describe its clinical efficacy also in cotext of COVID-19.

2) A large number of papers were published against use of hydroxychloroquine in COVID-19 patients, authors should consider to discuss this issue and add a paragraph in discussion to this point. 

3) Authors should consider a change in the title of article, e.g. changing "challenge" in title of article since they are not focusing on this issue.

Author Response

Reviewers 2.

Ragonnet et al describe the pharmacokinetics of hydroxychloroquine usage in ICU for COVID-19 patients. The study is interesting and well written. However, I shall suggest few points to discuss/change.

1)Authors only discuss the pharmacokinetics of different doses of hydroxychloroquine which is largely descriptive. I would suggest to describe its clinical efficacy also in context of COVID-19.

We are grateful to the reviewer for this relevant comment. The primary objective of this paper was to evaluate the pharmacokinetic variability of ICU patients and the impact on exposition to hydroxychloroquine and was then the major point discuss in the manuscript. As requested, clinical efficacy was also described in the discussion section. Please see the attachment.

2) A large number of papers were published against use of hydroxychloroquine in COVID-19 patients, authors should consider to discuss this issue and add a paragraph in discussion to this point. 

Although the primary objective of this paper was to evaluate the pharmacokinetic variability of HCQ in ICU patients, we agree with the reviewer and modification has been added in the end of the discussion section. Please see the attachment.

3) Authors should consider a change in the title of article, e.g. changing "challenge" in title of article since they are not focusing on this issue.

We agree with the reviewer and modification has been added in the manuscript. Please see the attachment.
